# The effect of a hiding box on stress levels and body weight in Dutch shelter cats; a randomized controlled trial

W. J. R. van der Leij[1☉¤a]*, L. D. A. M. Selman[2,3☉], J. C. M. Vernooij[4¤b], C. M. Vinke[3¤c]

1 Department of Clinical Sciences of Companion Animals, Shelter Medicine Programme, Faculty of Veterinary Medicine, Utrecht University, Utrecht, The Netherlands, 2 Department of Clinical Sciences of Companion Animals, Faculty of Veterinary Medicine, Utrecht University, Utrecht, The Netherlands, 3 Department of Animals in Science and Society, Faculty of Veterinary Medicine, Utrecht University, Utrecht, The Netherlands, 4 Department of Farm Animal Health, Faculty of Veterinary Medicine, Utrecht University, Utrecht, The Netherlands

☉ These authors contributed equally to this work.
¤a Current address: Department of Clinical Sciences of Companion Animals, Faculty of Veterinary Medicine, Utrecht University, Utrecht, The Netherlands
¤b Current address: Department of Farm Animal Health, Faculty of Veterinary Medicine, Utrecht University, Utrecht, The Netherlands
¤c Current address: Department of Animals in Science and Society, Faculty of Veterinary Medicine, Utrecht University, Utrecht, The Netherlands
* W.J.R.vanderleij@uu.nl

**Data Availability Statement:** All data files are available from the DataverseNL database (accession number: 10411/T8LKML) https://hdl.handle.net/10411/T8LKML.

## Abstract

While staying in an animal shelter, cats may suffer from chronic stress which impairs their health and welfare. Providing opportunities to hide can significantly reduce behavioural stress in cats, but confirmation with physical parameters is needed. Therefore, the aim of this study was to determine the effect of a hiding box on behavioural stress levels (scored by means of the Cat-Stress-Score) and a physical parameter, namely body weight, during the first 12 days in quarantine for cats newly arrived cats at a Dutch animal shelter. Twenty-three cats between 1 and 10 years of age were randomly divided between the experimental (N = 12) and control group (N = 11) with and without a hiding box. Stress levels were assessed on days 1, 2, 3, 5, 7, 9 and 12 according to the non-invasive Cat-Stress-Score (CSS). Body weights were measured on days 0, 7 and 12. Finally, adoption rates and length of stay (LOS) were determined. Major findings of the study are: (1) the mean Cat-Stress-Score decreased with time for all cats, but cats with a hiding box showed a significant faster decrease in the CSS, reaching a lower CSS-steady state seven days earlier than the control group; (2) nearly all cats in both groups lost significant body weight during the first two weeks; (3) hiding boxes did not significantly influence weight loss; (4) no differences were found in the adoption rates and the LOS between both groups. Hiding enrichment reduces behavioural stress in shelter cats during quarantine situations and can therefore be a relatively simple aid to shelter adaptation. It offers no prevention however against feline weight loss, which indicates a serious health risk for shelter cats.

**Funding:** W.J.R. van der Leij, the corresponding author of this manuscript, received financial support for this study from the Maria Naundorf Van Gorkum Fund (grant number: DG.023003) and from Ms. R. Brons. We thank them for their generous support of this research. The Maria Naundorf Van Gorkum Fund: https://www.uu.nl/en/organisation/alumni/well-spent/named-funds. The funders had no role in study design, data collection and analysis, decision to publish, or preparation of the manuscript.

**Competing interests:** The authors have declared that no competing interests exist.

## Introduction

Each year around 200 animal shelters in the Netherlands take in and rehome 27.000 stray and relinquished cats [1]. A shelter life is often associated with many stressors. Cats entering a shelter are introduced to a foreign environment with unfamiliar animals, people, sounds and smells. During these first days many of the cats struggle to adapt to these prolonged or repeated stressors and thus show stress responses [2–4]. Acute stress will encourage the animal to adapt to its surroundings. However, when prolonged aversive stimuli interfere with the adaptation ability, chronic stress will develop by causing a dysregulation of several major physiological systems like the hypothalamic–pituitary–adrenal (HPA) axis [5]. This may elicit clinical signs, such as hiding behaviour, defecating and urinating outside the litter box, decreased grooming or over-grooming behaviour and a loss of appetite [2,6–9]. Stress-induced long-term high cortisol levels can reduce the efficacy of the immune system against infectious diseases [1,6,7,9,10,11], and chronic stress can therefore harm a cat's health [6,8,12,13].

When in a state of stress, the majority of cats will stop eating. Tanaka et al. found that stress elicited a decrease in food intake, negatively correlated with stress scores [14,15]. This stress response can have grave impact on cats: severe body weight losses in only a short period of time can induce feline hepatic lipidosis [6,16,17].

Several studies show that stressed cats display increased alert resting behaviour behind their litter box in an environment without hiding opportunities [12,18,19]. This is interpreted as alternative hiding behaviour for it offers some concealment [12,18]. Real concealment can be offered by providing a hiding box to shelter cats. A study conducted by Kry and Casey [19] demonstrated a decrease in stress, measured by the Cat-Stress-Score (CSS), when shelter cats were offered hiding boxes. Weight loss during quarantine is another phenomenon in shelter cats associated with stress [14]. However, little research has been done on the preventive effect of a hiding box on this stress induced weight loss.

A previous study conducted by Vinke et al. [12] has been the first step in gathering scientific data about the effect of a hiding box on stress levels of newly arrived cats in a Dutch animal shelter during the first 14 days in quarantine situations. The results show that cats with a hiding box recovered at least 4 days earlier from stress than cats without a hiding box [12]. The present study was designed with more frequent CSS scoring between Day 5 and 12, to gain greater insight into the feline recovery from stress and to relate these behavioural stress levels to a physical parameter, such as body weight.

The primary aim of this study was to determine the effect of a hiding box on behavioural stress levels and on body weight of newly arrived cats in a Dutch animal shelter during the first 12 days in quarantine. The additional aim was to compare the Length of Stay (LOS) of cats in both study groups. It was hypothesized that a hiding box would significantly reduce stress levels of newly arrived cats compared to the non-hiding box group, reflected in a lower CSS, less weight loss and a shorter LOS.

## Materials and methods

The study was approved by the Animal Welfare Body Utrecht, after assessing the present study (Animal Welfare Body of Utrecht University, written declaration of 17 June 2016 by the Animal Welfare Officer). It was concluded that the study does not meet the definition of an animal experiment as defined in the Dutch Experiments on Animals Act and Directive 2010/63/EU because the animals encountered no discomfort.

## Animal shelter

This study was carried out at a Dutch animal shelter (Stichting Dierentehuis Arnhem en omstreken), a medium size animal shelter with an open intake of around 700 cats per year [20]. Cat housing is situated in five separate quarantine units, an isolation ward and an adoption unit, providing a maximum shelter capacity of 90 cats in total. Dutch legislation mandates that animal shelters must have a legal stray holding period of 14 days. Stray cats must be quarantined in solitary housing at intake and quarantine and isolation wards must be physically separated from the main shelter setting. Furthermore new cats must be vaccinated against FHV/FCV/FPV within five working days after intake [21]. The housing of the animals, their care and management provided by the shelter in this study is representative of the majority of Dutch shelters.

Informed consent was obtained from the shelter staff for this study. In order to relate this study to daily shelter management, the original shelter protocols about the intake of new animals, daily animal care and hygiene were generally accepted, and substantial adjustments were avoided.

**Animals.** For this study, 23 European short hair cats, 11 males and 12 females, were selected out of the cats entering the shelter between 4th November and 30th December 2015. Cats entering the shelter were examined at intake by the shelter staff for gender, breed and age and received a treatment against ecto- and endoparasites (Stronghold® and Milbemax®). Given that all the cats came in as strays, age was estimated in years. Within 5 days after intake, the shelter veterinarian performed a physical health check. During this veterinary check (during the morning hours), the cats were microchipped and vaccinated with an attenuated vaccine (Versifel CVR®) against feline panleukopenia virus (FPV), feline herpes virus (FHV-1) and feline calici virus (FCV). Intact cats were spayed or neutered after day 14.

Inclusion criteria for this study were based on breed (European shorthair), health status and age (between 1 and 10 years of age). When new cats showed no clinical signs of illness, obvious heat, pregnancy or signs of nursing during the physical examination at intake, they were included in this study. As it is not generally accepted practice in Dutch animal shelters to screen apparently healthy cats through diagnostic testing (e.g. FIV/FeLV) at shelter intake, apart from the physical examination, no additional information was available on the feline health status of the cats in this study.

All cats were observed for at least 12 days after intake.

Two cats participating in this study left the shelter before their last observation day: from the Hiding box group one cat went to a foster home, from the Control group one cat was released within a trap-neuter-release (TNR) programme. Data of both cats were excluded from this study. Two other cats were not included in data for the length of stay, but were included in data for the Cat-Stress-Score, body weight and the adoption rate. After the 12 days observation period, one of these cats (nr. 8, control group) proved to be infected with FeLV and was euthanized a few days after the quarantine period of 14 days, while another cat (nr. 19, control group), because of its semi-feral behaviour, was also released through the TNR programme. Given that shelters often take in these non-clinical, but infected, cats and stray cats being poorly socialized (and even rehome them), this study has included these two cats in three of the four measured parameters.

In this study, cats between approximately 1 and 10 years of age were included. The growth rate of cats younger than 1 year old [22] might interfere with the parameters of body weight used in this study, while the CSS of older cats (> 10 years) might be influenced by age related cognitive dysfunction [23].

As previous studies [19,24] found no gender-related significant differences in stress behaviour, both male and female cats were included in the present study. The 23 cats were randomly assigned to one of the two groups with and without access to a hiding box, using an online randomization tool [25,26].

**Housing conditions.** The cat housing in the two adjacent quarantine wards consisted of cages (L x W x H: either 84 x 95 x 80 cm with elevated perching shelves (at height of 28 cm) of 84 x 25 cm or 69 x 91 x 87 cm with shelves (equal height) of 69 x 25 cm) in which the cats were individually housed. Available floorspace per cage (cage floor plus perching shelf) was respectively 1.01 $m^2$ and 0.80 $m^2$.

Every cage was furnished with a food and water bowl, bedding of towels, and a litter box. The cages of the experimental group contained a hiding box that was placed at the right side at the back of the cage. To avoid place preference for towels as bedding, the towels covered the entire floor of the cage, including the shelf and the interior of the hiding box.

Cardboard boxes were used as hiding boxes and measured 44 x 31 x 26 cm (L x W x H). These boxes had two entrances (WxH 0.16 x 0.20 m) [12]. Hiding boxes were never reused.

Access to the cats in the quarantine wards was restricted to the caretakers and the observer. Natural daylight was provided through windows in both quarantine wards, combined with fluorescent lighting between 08:00 AM. and 5:00 PM. Daily temperatures in the quarantine wards ranged from 16.0 to 19.8 ˚C. In the quarantine wards, no dog vocalizations could be heard.

**Daily animal care.** The shelter staff cleaned the cages daily between 09:00 and 12:15 AM by removing waste and applying a spot-cleaning method [27]. During this procedure, cats remained in their cages. Litter boxes were cleaned daily with hot water and dried with clean paper towels. Cages were disinfected between cats or when indicated (e.g. diarrhea) with a chlorine disinfectant containing sodium dichloroisocyanurate (Halacid®).

Food was provided once daily between 9:30 and 10:00 AM and comprised of around 50g per day Adult Royal Canin® dry cat food (SC 365D) with a metabolizable energy content (ME) of 4066 kcal/kg (16.995 MJ/kg). Fresh water was provided ad libitum. Cats kept their own litter box for the duration of this study.

## Behavioural observations

Cats were given a habituation period of 24 hours after shelter intake (= Day 0), before behavioural assessment was performed [19]. Behavioural data were collected on days 1, 2, 3, 5, 7, 9 and 12 between 12:30 and 5:15 PM, during which interactions with caretakers were avoided.

Each cat was observed for 20 minutes per day by using video-recording. Outside the cage, a video camera (H.264 DVR) was mounted on a tripod at cage height. For new observations the combination camera-tripod had to be readjusted to the new cat cage. Video recordings were viewed in real-time in an adjacent room and stored for subsequent analysis (Fig 1). Only one camera was used for recording.

**Cat-Stress-Score (CSS).** Kessler and Turner [24,28] developed a 7-level Cat-Stress-Score (CSS), which has been used in several studies to estimate stress levels in confined cats [3,12,19,24]. This scoring system assesses the level of feline stress based on the posture of body elements (e.g. belly, legs, tail, head, eyes, pupils, ears, whiskers) and behaviour (vocalization and activity) as described in the ethogram of the UK Cat Behaviour Working Group [28]. The CSS ranges from 1 (fully relaxed) to 7 (terrorized).

One observer (LS) assessed the CSS score per cat on Day 1, 2, 3, 5, 7, 9 and 12. Intra-observer variation was minimized by observational training using (video) images of pre-described feline behaviours from previous experiments with shelter cats.

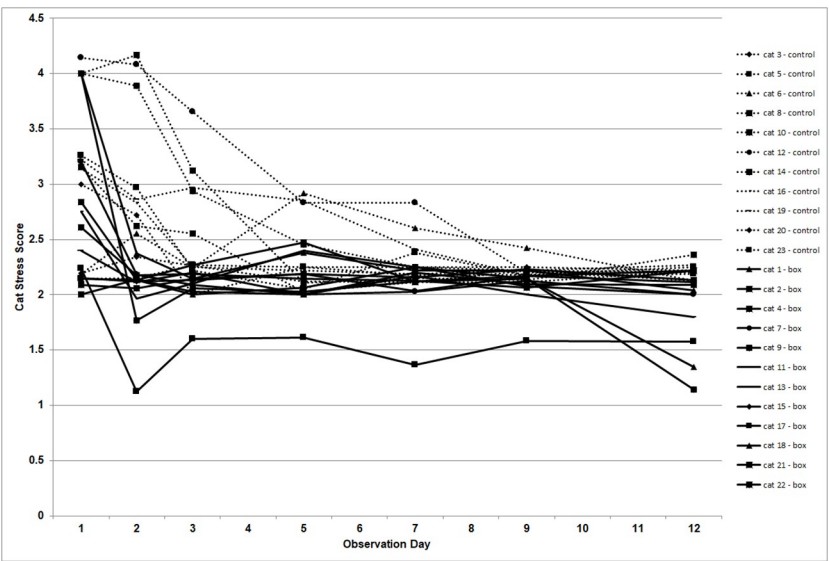

**Fig 1. Course of the Cat-Stress-Score in time of individual cats from the control group and the experimental group.** Line segments connect measurements within the same cat to show the change of CSS in course of time. Dotted lines: individual cats without hiding boxes (control group). Solid lines: individual cats with hiding box (experimental group).

After the video camera had been positioned, the scan sampling started after 2 minutes during which the cat habituated to the novel situation. The cat was subsequently scored according to the Scan Sampling method, in which four scores (= four samplings) were made during the observation time (the 1st observation at 5 min, the 2nd at 10 min, the 3rd at 15 min and the 4th at 20 min) [29]. Imperceptible posture and behavioural elements were noted as missing values.

Each of the elements of the Cat-Stress-Score was scored separately. The scores of the four samplings were averaged to assign an overall CSS for each separate cat per day.

## Body weight

During the study every cat was weighed on Day 0, 7 and 12 by using an electronic scale (accuracy ± 10 g). The standardized shelter feeding regime consisted of approximately 50g per cat per day of Adult Royal Canin (RC)® dry cat food, which equals 203.3 kcal or 849.8 kJ per cat per day. All cats were offered the same diet, with some individual temporal exceptions, which are mentioned in the text (oral medication would be given mixed with tinned food). The shelters' feeding regime did not include the monitoring of the daily intake of food per cat and neither did our study design.

To secure adequate nutrition for the cats in this study, the daily caloric feline requirements (FEDIAF guidelines (80 kcal (335 kJ) ME per $kg^{0.67}$)) were determined per individual cat [30].

## Adoption rates and length of stay (LOS)

In order to determine the effect of a hiding box in quarantine situations on the subsequent adoption success, the adoption dates of the cats in this study were noted. Adoption rates (= # cats adopted / all cats in this study) and the length of stay (LOS: number of days between the shelter intake of the cats in this study and its day of adoption) was determined per cat. The LOS included the mandatory quarantine period of two weeks and only included adopted cats,

excluding cats that were euthanized or returned to their outdoor environment after finishing this study.

## Statistical analyses

A randomised controlled trial (RCT) design was used [31,32]. Data were stored in Microsoft Excel 2010 files (Microsoft Corp, Redmond, Wash.). Two statistical software programs were used for analysis of the data:

- SPSS (IBM Corp, Armonk, NY version 25) for the two-sample T-test and chi-square test.

- R (version 3.3.0) for the linear mixed regression models [33].

For the statistical analysis of effect of time and hiding box on the CSS (model 'CSS-Time-Box') a linear mixed regression model [34] was assumed, with the CSS as the outcome, while Time after arrival, the availability of a hiding box and the interaction between both were used as explanatory factors. CatID was used as the random effect to take the correlation between observations within cat into account. An AR1 correlation between the time points was added as well as a variance model to allow different variances for the separate time points. A maximum likelihood-based method was used to calculate the Akaike's Information Criterion (AIC) to select the best model using a backward selection approach (smaller is better).

For the statistical analysis of effect of time and hiding box on the body weight, a linear mixed regression model [34] was used to analyse the weight as the outcome and Time after arrival, the availability of a hiding box and the interaction between both as explanatory factors. Although keeping the box in the linear mixed model resulted in a worse fit of the model, the availability of the box nevertheless was added in coherence with our primary aim. Also, in this model CatID was used for the random effect.

The validity of both models was confirmed by a visual inspection of the residuals for normality and constant variance.

Per experimental group the number of adopted cats was analysed using the chi-square test, while the length of stay (LOS) was analysed using the two-sample T-test. The assumptions for these variables for equal variance (Levene's test) and for normal distribution (Shapiro-Wilk's test) were met.

We reported the estimated effects of the availability of a hiding box according the reporting guidelines for randomized controlled trials (www.reflect-statement.org).

## Results

### Characteristics of the study population

The experimental group consisted of 12 cats (6 males and 6 females) of which the estimated age ranged between 1 and 7 years (mean: 3.3 years, SD: 2.2). The control group consisted of 11 cats (5 males and 6 females) with estimated ages between 1 to 10 years (mean: 4.9 years, SD: 3.1, with n = 10: due to her semi-feral behaviour no age could be estimated of cat nr 19). At intake, the control cats were on average 300 grams heavier than those in the experimental group.

The cats in this study are presented in the appendix with their ID, experimental group, gender, age, bodyweight at intake (kg) and the quarantine wards they went to after intake.

### Daily Cat-Stress-Score (CSS): Behavioural assessment

The time-dependent reduction of the individual CSS per cat in both groups is visualized in Fig 1.

**Table 1. Results of the model for the Cat-Stress-Score with 95% confidence interval, influenced by day and availability of a hiding box and interaction between both.**

| Time | Cat-Stress-Score (CSS) | | | |
|---|---|---|---|---|
| | Control | | Hiding box | |
| days after Intake | Estimated mean | 95% CI[1] | Estimated mean | 95% CI[1] |
| Day 1 | 3.13[2] | 2.74–+3.53 | -0.43[4] | -0.97–+0.12 |
| Day 2* | -0.11[3] | -0.43–+0.20 | -0.99[4] | -1.38–-0.61 |
| Day 3* | -0.54[3] | -0.91–-0.17 | -0.51[4] | -0.79–-0.23 |
| Day 5* | -0.76[3] | -1.15–-0.37 | -0.25[4] | -0.47–-0.03 |
| Day 7* | -0.82[3] | -1.21–-0.42 | -0.23[4] | -0.40–-0.05 |
| Day 9* | -0.92[3] | -1.32–-0.53 | -0.12[4] | -0.24–-0.01 |
| Day 12* | -0.91[3] | -1.34–-0.49 | -0.33[4] | -0.57–-0.08 |

[1] CI = Confidence Interval

[2] Estimated mean CSS in cats in Control group on Day 1.

[3] Estimated difference between mean CSS at specified day in Control group compared to mean CSS on Day 1 of same cats.

[4] Estimated difference between mean CSS at specified day in cats of group with Hiding box compared to mean CSS of cats in group Control group at same day.

* Significant difference between Mean CSS of the hiding box group and the mean CSS in the control group of the same day.

Cats from the hiding box group reached a steady state sooner (on Day 2) than cats from the control group (on Day 9). The model results for the mean CSS are presented in Table 1. The estimated means of the CSS of the hiding box group (mean CSS = 2.7) and the control (mean CSS = 3.1) at Day 1 are similar as their difference is not significant (-0.4, 95% CI:-0.97 to +0.12). At all other days the mean CSS of the hiding box group is significantly lower than the mean CSS in the control group, largest at Day 2 (-0.99, 95%CI: -1.38 to -0.61) and decreasing in difference between the groups on Day 12 (-0.33, 95%CI: -0.57 to -0.08).

## Body weight

For the comparison of both experimental groups, the absolute body weight was used. The initial weight difference of 300 grams between both groups reduced to 210 grams at Day 7 and Day 12. Cats in the control group lost overall 7.7% of their initial body weight, while cats with a hiding box lost 6.3% of their initial body weight during those 12 days (Table 2). The initial weight and weight reduction between the groups, however, proved not to be significant.

**Table 2. Results of the model for body weight with a 95% confidence interval, influenced by day and availability of a hiding box and interaction between both.**

| Time | Body weight | | | | | |
|---|---|---|---|---|---|---|
| | Control | | | Hiding Box | | |
| Days after Intake | % change from Day 0 | Estimated mean (kg) | 95% CI[1] | % change from Day 0 | Estimated mean (kg) | 95% CI[1] |
| Day 0 | 0 | 4.39[3] | 3.77–5.01 | 0 | -0.30[5] | -1.16–0.56 |
| Day 7 | -6.1 (SD[2] 4.1) | -0.25[4] | -0.35–-0.15 | -4.5 (SD[2] 3.9) | -0.21[5] | -1.07–0.65 |
| Day 12 | -7.7 (SD[2] 5.1) | -0.32[4] | -0.42–-0.22 | -6.3 (SD[2] 4.4) | -0.21[5] | -1.07–0.65 |

[1] CI = Confidence Interval

[2] SD = Standard deviation of the % change in body weight compared with Day 0.

[3] Estimated mean Body weight of cats in the Control group on Day 0.

[4] Estimated difference between the mean Body weight of cats in the Control group on the specified Day compared to the mean Body weight of the same cats on Day 0.

[5] Estimated difference between the mean Body weight of cats in the Hiding box group and the mean body weight of cats in the Control group on the specified Day.

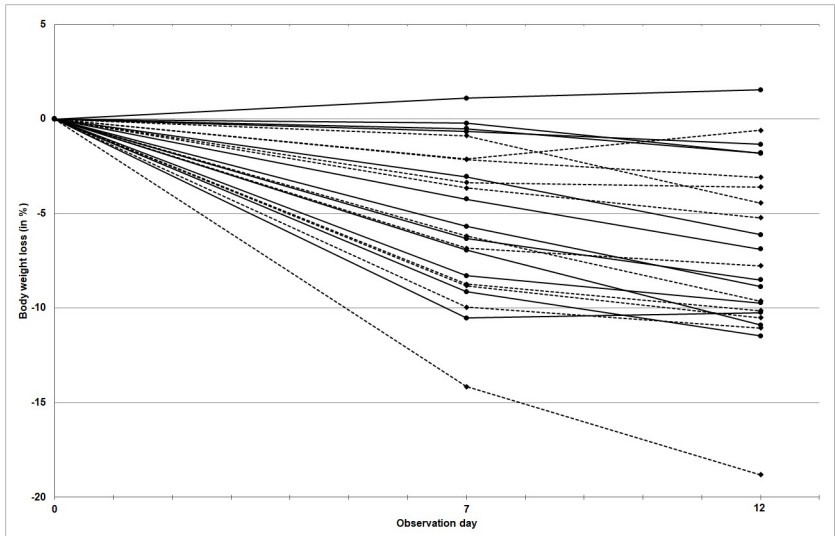

**Fig 2. The proportional change (%) in body weight in individual cats from the control group and the experimental group.** Line segments connect measurements within the same cat to show the change of body weight in course of time. Dotted lines: individual cats without hiding boxes (control group). Solid lines: individual cats with hiding box (experimental group).

The individual proportional decrease in body weight is visualized in Fig 2. All cats except one lost weight during both weeks. When weight loss at Day 12 was calculated as a percentage of initial body weight at intake, it was found that 7 of the 23 (35%) cats lost ≤ 5% of their body weight, whereas 15 of the 23 (65%) cats lost 5% or more of their weight. The maximum body weight loss was found in cat nr. 8 (control group), which lost 19% of its initial weight in 12 days and was diagnosed with an infection of FeLV a few days after completing this study.

## Adoption rates and length of stay (LOS)

Of the 23 shelter cats in this study, 21 were rehomed after the observation period was completed. In the control group, 9 out of 11 cats were adopted (82%), in the experimental group 12 out of 12 (100%). No significant difference was found in the adoption rate between the two groups (p = 0.55).

As we defined LOS as the number of days between the shelter intake of a cat and its day of adoption, 2 cats were not included in this data set, for they were not adopted.

The mean LOS for the control group (n = 9) was 24.1 days (SD 5.4, range 15–30 days) and for the hiding box group (n = 12) was 22.9 days (SD = 4.4, range 16–30 days). No difference in the mean LOS was found between the control and the hiding box group (p-value = 0.58).

## Discussion

The aim of the present study was to determine the effect of a hiding box on behavioural stress levels and body weight in shelter cats during the first 12 days of quarantine. While in a previous study cats were monitored on days 1 through 5 and the 14[th] day [12], this new study added more insight about the differences in CSS between Day 5 and Day 12.

The most important findings of this study are:

- The mean Cat-Stress-Score decreased with time for all cats, but cats with a hiding box showed a significant faster decrease in the CSS and recovered from stress seven days earlier than the control group.

- Nearly all cats lost significant body weight during the first two weeks. On average, cats with hiding boxes lost 40 grams less of their initial body weight compared with cats without a box, although this difference was not significant.

- The mean adoption rates and the LOS of cats with and without hiding boxes were equal.

The cages used for the cats in this experiment, varied in dimensions: the smaller cage offered 80% of the floor space of the larger one. Although Kessler and Turner [35] showed an effect of cage dimensions (0.49 m$^2$ versus 1.02 m$^2$) on stress levels in individual cats, they considered other qualitative aspects of the housing environment to be of importance as well. More recent research [36] studied the effect of cage dimensions (1.1 m$^2$ versus 0.56 m$^2$) combined with different feline care regimes. Doubling the cage sizes did not influence the pattern of feline responses, suggesting that physical space solely may be of less importance to cats, in contrast to managed and enriched environments that did significantly affect the cats. Therefore, in this present study, the effects of hiding enrichment were studied in spite of a minor size deviation between the individual cat cages.

## Cat-Stress-Score (CSS): Behavioural assessment validation

In this study, cats with a hiding box showed a significantly faster decrease of behavioural stress compared to the control group, which was most prominent during the first observation days. These results were in line with a study of Gourkow and Fraser, in which the mean CSS of cats, housed in single barren cages without positive human-cat interaction, was higher compared to the other groups and only reached a similar CSS on Day 9 [15].

The findings of the present study complete the results obtained by Vinke et al., where the hiding box group recovered at least four days earlier. By increasing the number of observational days during the first 12 days, the current research provides more details about reaching the CSS-steady state, indicating that hiding boxes accelerate the recovery of behavioural stress by seven days. The hiding box clearly helps the shelter cat to adapt more quickly to a stressful new environment thus preventing the development of chronic stress [19].

## Body weight

This study shows a significant decrease in feline body weight during the first 12 days in an animal shelter. Approximately a third of the cats lost less than 5% body weight during the first 12 days, while two-third lost over 5%. These results agree with previous findings of Tanaka et al., in which 57% of their cat population showed 5% or more weight loss during their shelter stay [14]. Stressed cats are more likely to lose body weight. Cats in shelters [14], in laboratories [18], in boarding facilities [unpublished data] and even privately-owned cats [6] display this general stress response in challenging situations. However, when otherwise healthy cats lose weight unintentionally, it is a dramatic indicator of a health risk.

Weight loss can be caused by insufficient nutritional management (the shelter offers inadequate quantity and/or quality of food) and also by a decrease in feline appetite due to a physical stress response. Although food intake was not registered in the present study, it was observed that some cats were completely anorectic, especially during the first days. For the shelter this was the reason to standardize the feeding schedule of 50 g dry cat food per cat per day. According to the FEDIAF guidelines [30] for daily caloric feline requirements, during this study cats

over 4.01 kg might have been offered an inadequate amount of food. With an individual requirement of 80 kcal (335 kJ) ME per $kg^{0.67}$, 50 g dry cat food per day will meet maintenance energy requirements of cats up to a body weight of 4.01 kg. Cats weighing over 4.01 kg, need more Adult RC food daily. Of the 23 cats, 13 (57%) cats weighed more than 4.01 kg. The heaviest cat weighed 6.41 kg at intake and hence required at least 68.4 grams of cat food per day. During the daily observation, however, cats rarely finished their food rations during these first two weeks. An inadequate quantity of food was therefore not considered to be the cause of the observed body weight loss.

The effect of stress on the body weight of shelter cats was first shown by Tanaka et al. [14], who found a negative correlation between food intake and stress scores of cats. The conclusion was that cats admitted to an animal shelter were likely to lose weight while in the shelter. These results are consistent with our findings, indicating that a decrease of feline appetite caused by a physical stress response, is most likely responsible for the weight loss.

Although the provided commercial food in this study was of a high quality, there is less understanding of the role of palatability of food for shelter cats in relation to weight loss. The only cat in this study to gain weight (cat 22, experimental group) received medication for diarrhea (fenbendazole 50 mg/kg, PO, q 24 h) mixed with canned food. This gives an indication of the importance of palatability of food for shelter cats.

Although the analysis of the effect of time and the presence of a hiding box on the body weight suggested that there was a difference between the two groups in body weight losses, as cats with hiding boxes showed approximately 40 grams less weight loss compared to the control group, this difference was not significant. For the individual cat, however, this could be biologically relevant, for weight loss due to feline anorexia has a serious impact on a cat's health, increasing the risks of hepatic steatosis [6,16,17]. Therefore, more research is necessary to monitor these cats for a longer period of time, to register the process of adaptation to the new environment in correlation to the weight losses and to experiment with ways of preventing or reducing body weight losses in shelter cats.

Apart from stress, progressive weight loss can also be a sign of serious medical problems [37]. One of the cats from the control group showed a weight loss close to 20% in 12 days and was eventually diagnosed with FeLV. Shelters could use weight loss during quarantine time as an early warning sign for serious declines in physical conditions, but this would require weighing as a standard monitoring procedure.

## Adoption rates and length of stay (LOS)

Providing cats with hiding enrichment at any stage of their shelter stay requires an investment in shelters' scarce time and money. Shelter staff sometime have their reservations about using hiding boxes, for it might decrease the visibility of cats to potential adopters and therefore slow down adoption rates (personal communications). Although Kry and Casey found no significant difference in the Length of Stay between cats with and without hiding boxes in the adoption ward [19], in the present study the hiding enrichment itself could not have influenced the adopters' choices based on the (in)visibility of the cat. Hiding boxes were only present in 12 of the 23 cages during the first 12 days of quarantine time, while no hiding boxes were available in the adoption area.

Our study showed a decrease in the CSS during quarantine time, while eventually both groups had similar rates for adoption and LOS. In sum, providing hiding enrichment to shelter cats benefits the welfare of the animals without having negative consequences for the shelter (like an increased LOS per cat).

## Finally, stressors versus signals of safety?

While this study proved again a significant decrease of the behavioural stress response when shelter cats were offered a hiding opportunity, the effects on body weight were minor. These results challenge our point of focus on stress in shelter animals: a shelter environment offers numerous stressors for which feline hiding behaviour appears not always sufficient enough to induce adaptation within the first 12 days after intake. New theories on human stress response mechanisms might shed some light on the feline stress response in these complex shelter environments and contribute to more practical tools for stress reduction. According to Brosschot [38], who introduced the Generalized Unsafety Theory of Stress (GUTS), 'the stress response of the body is always "on" and it stays on as long as there is no obvious safety.' This default response can only be inhibited when 'signals of safety' are perceived by the animal. We therefore should not look for the causation of a stress response, but rather ask ourselves 'what stops the stress response?'. When present results are reviewed in the light of this GUTS, the hiding enrichment itself caused a decrease in feline behavioural stress scores, but did not provide an adequate signal of safety (SOS) to prevent weight loss in most cats. This GUTS approach asks for a comparison of the effect of distinct SOSs (such as hiding materials, food presentations, enriched feeding, feline pheromones, human contact, increased cage space, solitary housing, etc) and for the reinforcing effects of combining these signals on the majority of shelter cats. In addition to focusing on reduction of numerous stressors in the shelter environment, we should also search for SOSs that are strong enough to inhibit the stress response and thus create a situation which the majority of animals can perceive as safe.

## Limitations

Conducting the present randomized controlled trial research in an operating animal shelter comes with its inherent limitations. Restrictions in research time consequently affects the number of animals available, shelter management does not always coincide with the formation of experimental groups in identical housing while shelter protocols for animal care need to be respected.

This study is based on a small sample of shelter cats and only captures the first 12 days after intake. The evaluation of stress impacting the body weight of cats likely requires a longer assessment period and a larger dataset to be able to apply these insights to a wider cat population in shelters. Despite these limitations, the current findings provide evidence that hiding boxes may not be able to prevent significantly weight loss in shelter cats and are instructive for our understanding of the effects of hiding enrichment on the feline behavioural parameters.

During this research, feeding protocols, based on long-term experience of the shelter staff, were unaltered. By doing so this study created a better understanding of the effect of shelter feline care on behavioural parameters and body weight of these cats. For subsequent research, however, it is recommended to gather data on the daily intake of food and water per individual shelter cat.

The behavioural stress parameter of the Cat-Stress-Score was combined with the physical parameter of body weight. Although stress has previously been shown to negatively affect feline body weight [14], the inclusion of a specific biochemical marker of stress would have strengthened the outcomes of this study. Preferably both physical and biochemical feline stress markers are combined in future studies.

The shelter cats were observed with a video camera mounted in front of their cages, a technique used in several studies [36,39,40,41]. In the previous [12] and current study cats were given a habituation time to positioning the camera of 2 minutes, before starting the observation. To the authors' knowledge the stress inducing effect of placing a novel static object in

front of a cat cage has never been studied in cats. Its effect on the loss of body weight in both groups cannot be ruled out and therefore new research is recommended to investigate the stress inducing effect of this observation method in shelter cats.

## Conclusion

Providing hiding boxes can be a relatively simple way for cats to self-manage stress and to adapt faster to the shelter environment. The majority of the shelter cats, however, lost (considerable) weight during the quarantine period in an animal shelter. Providing them with hiding enrichment during that period, does not prevent weight loss. Nor do hiding boxes have effect on the adoption rates and the length of stay of both groups.

However, instead of keeping focus on identifying and reducing stressors in a very challenging environment like an animal shelter, an additional approach could be found in the application of 'signals of safety' (SOS), strong enough to attenuate or even inhibit the stress response and thus create a situation that animals can perceive as safe.

## Supporting information

**S1 Appendix. Diagram of the experimental set up, observer and both camera position in the quarantine wards in the animal shelter.**
(BMP)

**S2 Appendix. Baseline characteristics of treatment cohorts in a randomized field trial comparing quarantine cat housing with and without hiding opportunities in one Dutch animal shelter.**
(BMP)

## Acknowledgments

We thank the staff of the animal shelter 'Dierentehuis Arnhem en omstreken' for access to their facilities and their invaluable support during this research. The authors thank the Dutch Society for the Protection of Animals for providing the hiding boxes used in this study.

We wish to thank Prof. Dr. J.W. Hesselink and Dr. R.J. Corbee for their assistance with this study and manuscript preparation.

## Author Contributions

**Conceptualization:** W. J. R. van der Leij, C. M. Vinke.

**Data curation:** W. J. R. van der Leij, L. D. A. M. Selman, J. C. M. Vernooij.

**Formal analysis:** W. J. R. van der Leij, L. D. A. M. Selman, J. C. M. Vernooij.

**Funding acquisition:** W. J. R. van der Leij, C. M. Vinke.

**Investigation:** W. J. R. van der Leij, L. D. A. M. Selman.

**Methodology:** W. J. R. van der Leij, L. D. A. M. Selman, C. M. Vinke.

**Project administration:** W. J. R. van der Leij.

**Resources:** W. J. R. van der Leij.

**Software:** J. C. M. Vernooij.

**Supervision:** W. J. R. van der Leij, C. M. Vinke.

**Visualization:** W. J. R. van der Leij, L. D. A. M. Selman, J. C. M. Vernooij.

**Writing – original draft:** W. J. R. van der Leij, L. D. A. M. Selman.

**Writing – review & editing:** W. J. R. van der Leij, J. C. M. Vernooij, C. M. Vinke.

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
