## [Decision Letter · Decision Letter 0]

21 Jul 2019

PONE-D-19-15024

The effect of a hiding box on stress levels and body weight in Dutch shelter cats; a randomized controlled trial

PLOS ONE

Dear Mrs. van der Leij,

Thank you for submitting your manuscript to PLOS ONE. After careful consideration, we feel that it has merit but does not fully meet PLOS ONE’s publication criteria as it currently stands. Therefore, we invite you to submit a revised version of the manuscript that addresses the points raised during the review process.

We would appreciate receiving your revised manuscript by Sep 04 2019 11:59PM. To enhance the reproducibility of your results, we recommend that if applicable you deposit your laboratory protocols in protocols.io, where a protocol can be assigned its own identifier (DOI) such that it can be cited independently in the future. For instructions see: http://journals.plos.org/plosone/s/submission-guidelines#loc-laboratory-protocols

We look forward to receiving your revised manuscript.

Kind regards,

Juan J Loor

Academic Editor

PLOS ONE

Journal Requirements:

Reviewers' comments:

Reviewer's Responses to Questions

**Comments to the Author**

1. Is the manuscript technically sound, and do the data support the conclusions?

Reviewer #1: Partly

Reviewer #2: Partly

Reviewer #3: Yes

Reviewer #4: Yes

2. Has the statistical analysis been performed appropriately and rigorously? 

Reviewer #1: Yes

Reviewer #2: I Don't Know

Reviewer #3: Yes

Reviewer #4: Yes

3. Have the authors made all data underlying the findings in their manuscript fully available?

Reviewer #1: Yes

Reviewer #2: Yes

Reviewer #3: Yes

Reviewer #4: Yes

4. Is the manuscript presented in an intelligible fashion and written in standard English?

Reviewer #1: Yes

Reviewer #2: Yes

Reviewer #3: Yes

Reviewer #4: No

5. Review Comments to the Author

Reviewer #1: The study confirms previous findings that a hiding box can reduce behavioral indicators of stress in shelter cats. It also attempted to determine if this behavioral change decreased the expected initial weight loss seen in shelter cats. It is not clear that enough animals were evaluated or that the amount of food ingested by each cat was measured. It was clear that not all cats had the same diet (one cat that gained weight was given wet food). So this goal was not reached by the study. Finally, the length of stay of the cats was compared. However, hiding boxes were not used in the adoption area. So the reasoning for how hiding boxes affect length of stay is not clear. Did the authors predict that the cats would be less social when they were removed from the "safety" of the box and transferred to the adoption area?

The study is relatively well-written with a few grammatical errors. However, the discussion should be shortened significantly.

Specific suggestions: Address the concern of low power (low N) for the statistics involving weight loss.

Lines 37-38: Grammar suggestion "...but cats with a hiding box showed a significantly faster..."

Line 101: Spelling "vaccine"

Line 110: Grammar suggestion "Two cats participating in this study left..."

Figure 1 is unnecessary.

Lines 187-189. Clarify the calculation of LOS. Which cats? All cats in the shelter? Only cats in the study?

Lines 222-228: The difference observed in the initial average weight of the groups would be beneficial to include here.

Lines 230-183: Report what the individual cats ate. Did they all finish the food they were given each day?

Shorten the discussion. A great deal of information provided is repetitious. Clarify what is known and what is presumed/thought/supposed.

Measurement of a specific biochemical marker of stress would have been helpful to report.

Reviewer #2: This is a relatively good study, however the housing used for cats in the study was not the same across the 2 groups. Housing may affect stress levels in individual cats and the differences in the size of the housing used looks to be significant - one cage type provided 1221cm squared of floor space and the other was 945cm squared of floor space. It was not indicated that this was controlled for in the study. Of the 12 cats housed with hiding structures 10 were in the larger housing units. The control group was more evenly distributed between small and larger cages with 6 controls in the smaller cage and 5 in the bigger cage. I am concerned this is confounding variable in this study and should be controlled for.

Reviewer #3: Reviewer report for PONE-D-19-15024

This interesting and well-written study that tested the hypothesis that the introduction of a hiding box for cats will decrease their stress. The authors approached the hypothesis by testing it in a shelter environment during the quarantine period, where the level of stress of newly incoming cats is high. The authors found that by providing a hiding box, cats achieved lower stress levels significantly earlier than control cats which did not have a hiding box. I think that the study is very important and its results critical for establishing new regulations for the management of stress levels in shelters across the world. Hence, I strongly support its publications.

Below please find my specific comments:

Lines 50-54: What is the definition of ‘chronic’? I think it will benefit your manuscript if you clearly define a ‘window of time’ that directly correlates with well-defined clinical outcomes.

Line 66: The reference number for Vinke et al., is missing.

Line 79: Because the Ethics committee approved the study, then please provide the protocol number.

Line 102: ‘day 14’ not ‘Day 14’.

Lines 120-122: The structure of this sentence is unclear. It needs to be rewritten.

Line 124: Your manuscript will benefit from clearly defining the method of randomization. Was according to an online randomization tool? Coin toss? Etc.

Line 138: Do you think that the shelter in this study represents the setting in which many other shelters in the Netherlands and worldwide are set-up? In other words, do you know whether many / most other places have cats mixed with dogs and other stressors such as large traffic within the shelter that were not present in this shelter? You are attempting to test a method for stress prevention/reduction which is great, but if the setting of the study may not represent many other facilities, then the external validity of this study is in question. I am writing this from the standpoint of a reviewer who supports the publication of this study, so I am not looking for a defensive response… I am suggesting that you carefully think about this point and that in the revised manuscript you will discuss it and also point out how the results of this study could be leveraged to promote new regulation for shelter management and could be used as a stepping board for upcoming studies.

Lines 150-153: Are there any previous studies that indicated that the introduction of a new static object in front of a cat cage is not a stressful situation by itself. If there are studies that have shown it is not, then please cite them. Otherwise, in the limitation paragraph among the other things that I pointed out, you will need to discuss the possibility that mounting a camera in front of the cage could have induced stress and in theory, could be partially the reason for why there was no significant difference in body weight between the groups. Also, if it was not explored before, you should mention that it is something that needs to be studied in the future.

Line 170-171: This is related to my comment for lines 150-153; how do we know that 2 minutes is enough time. How do you know that it does not lead to prolonged stress…? If there is previous research to support that it is not, then cite it and it will make your study stronger. If you do not know, you will need to discuss it under the limitation paragraph.

Line 213: What did you mean when you used the word ‘constance’? is this a typo?

Figure 2: I strongly suggest that you will consider plotting the figure as a series of grouped boxplots with the CSS on the y-axis and the days it was assessed on the x-axis. Firstly, I think that it will be clearer than a bunch of lines that intersect and overlap with each other. Secondly, the CSS is not a continuum and therefore should not have a connecting line. For example, you do not know if there wasn’t a peak of stress in one of the days when it was not evaluated or during other hours on the days that you did…

Table 1: I strongly suggest that you will consider reconstructing your table such that the left column will have the days; the next column will have the control mean (95CI), and in the last column the hiding box mean (95CI).

Table 2: I strongly suggest that you will consider reconstructing your table such that the left column will have the days; the next column will have the control mean BW change from baseline (95CI) and in the last column the hiding box mean BW change from baseline (95CI). It would be much more meaningful for the reader to know the %change in BW from baseline. You can rerun your regression by incorporating it into your data file as a new column with the percent change for days 7 and 12 relative to baseline.

Figure 3: I strongly suggest that you will consider plotting the figure as a series of grouped boxplots with the %change in body weight on the y-axis and the days it was assessed on the x-axis — same reasons as for figure 1.

Line 298: please delete the word ‘however.’

Line 403: the limitation paragraph should include the limitations I have discussed throughout my reviewer response. In addition, you need to address the limitation that the CSS in this study was not coupled with biomarkers of stress, for example, cortisol.

Reviewer #4: This is an interesting and scientifically sound study that does contribute novel information in a research space increasingly convincing us that cats like boxes. Review the manuscript with a close eye to correct choice of words - getting it proofread by a third party may be helpful here. Examples include e.g. loose instead of lose; expresses instead of expressed.

6. PLOS authors have the option to publish the peer review history of their article (what does this mean?). If published, this will include your full peer review and any attached files.

Reviewer #1: No

Reviewer #2: No

Reviewer #3: No

Reviewer #4: No

---

## [Author Response · Author response to Decision Letter 0]

27 Aug 2019

Please see attached Word file: Response to reviewers PONE-D-19-15024

---

## [Decision Letter · Decision Letter 1]

24 Sep 2019

The effect of a hiding box on stress levels and body weight in Dutch shelter cats; a randomized controlled trial

PONE-D-19-15024R1

Dear Dr. van der Leij,

We are pleased to inform you that your manuscript has been judged scientifically suitable for publication and will be formally accepted for publication once it complies with all outstanding technical requirements.

With kind regards,

Juan J Loor

Academic Editor

PLOS ONE

Additional Editor Comments (optional):

Reviewers' comments:

Reviewer's Responses to Questions

**Comments to the Author**

1. If the authors have adequately addressed your comments raised in a previous round of review and you feel that this manuscript is now acceptable for publication, you may indicate that here to bypass the “Comments to the Author” section, enter your conflict of interest statement in the “Confidential to Editor” section, and submit your "Accept" recommendation.

Reviewer #1: (No Response)

Reviewer #3: All comments have been addressed

Reviewer #4: All comments have been addressed

2. Is the manuscript technically sound, and do the data support the conclusions?

Reviewer #1: No

Reviewer #3: Yes

Reviewer #4: Yes

3. Has the statistical analysis been performed appropriately and rigorously? 

Reviewer #1: Yes

Reviewer #3: Yes

Reviewer #4: Yes

4. Have the authors made all data underlying the findings in their manuscript fully available?

Reviewer #1: Yes

Reviewer #3: Yes

Reviewer #4: Yes

5. Is the manuscript presented in an intelligible fashion and written in standard English?

Reviewer #1: Yes

Reviewer #3: Yes

Reviewer #4: Yes

6. Review Comments to the Author

Reviewer #1: I'm really sorry, but I just don't feel the study was controlled enough to be sure confounding factors (diet, initial weight, cage size, etc) didn't affect outcome. I recommend repeating the study and considering these factors before publishing any conclusions.

Reviewer #3: I am pleased with the revision that the authors made. I would like to thank the authors for their hard work.

Reviewer #4: The authors have successfully integrated reviewer feedback and strengthened their original submission.

7. PLOS authors have the option to publish the peer review history of their article (what does this mean?). If published, this will include your full peer review and any attached files.

Reviewer #1: No

Reviewer #3: No

Reviewer #4: No

---

## [Editor Report · Acceptance letter]

7 Oct 2019

PONE-D-19-15024R1 

The effect of a hiding box on stress levels and body weight in Dutch shelter cats; a randomized controlled trial 

Dear Dr. van der Leij:

I am pleased to inform you that your manuscript has been deemed suitable for publication in PLOS ONE. Congratulations! Your manuscript is now with our production department. 

With kind regards,

on behalf of

Dr. Juan J Loor 

Academic Editor

PLOS ONE